# Extracellular Nanovesicles Secreted by Human Osteosarcoma Cells Promote Angiogenesis

**DOI:** 10.3390/cancers11060779

**Published:** 2019-06-05

**Authors:** Francesca Perut, Laura Roncuzzi, Nicoletta Zini, Annamaria Massa, Nicola Baldini

**Affiliations:** 1Laboratory for Orthopaedic Pathophysiology and Regenerative Medicine, IRCCS Istituto Ortopedico Rizzoli, 40136 Bologna, Italy; laura.roncuzzi@ior.it (L.R.); annamaria.massa88@gmail.com (A.M.); 2CNR-National Research Council of Italy, Institute of Molecular Genetics, 40136 Bologna, Italy; nicoletta.zini@cnr.it; 3IRCCS Istituto Ortopedico Rizzoli, 40136 Bologna, Italy; 4Department of Biomedical and Neuromotor Sciences, University of Bologna, 40123 Bologna, Italy

**Keywords:** osteosarcoma, extracellular nanovesicles, angiogenesis, acidity

## Abstract

Angiogenesis involves a number of different players among which extracellular nanovesicles (EVs) have recently been proposed as an efficient cargo of pro-angiogenic mediators. Angiogenesis plays a key role in osteosarcoma (OS) development and progression. Acidity is a hallmark of malignancy in a variety of cancers, including sarcomas, as a result of an increased energetic metabolism. The aim of this study was to investigate the role of EVs derived from osteosarcoma cells on angiogenesis and whether extracellular acidity, generated by tumor metabolism, could influence EVs activity. For this purpose, we purified and characterized EVs from OS cells maintained at either acidic or neutral pH. The ability of EVs to induce angiogenesis was assessed in vitro by endothelial cell tube formation and in vivo using chicken chorioallantoic membrane. Our findings demonstrated that EVs derived from osteosarcoma cells maintained either in acidic or neutral conditions induced angiogenesis. The results showed that miRNA and protein content of EVs cargo are correlated with pro-angiogenic activity and this activity is increased by the acidity of tumor microenvironment. This study provides evidence that EVs released by human osteosarcoma cells act as carriers of active angiogenic stimuli that are able to promote endothelial cell functions relevant to angiogenesis.

## 1. Introduction

Angiogenesis is a pivotal process in osteosarcoma (OS) development and progression [1,2]. Accordingly, several anti-angiogenic compounds have been considered in preclinical experiments and clinical trials in an attempt to improve the dismal prognosis of high-grade OS patients. The majority of these drugs are designed to interfere with vascular endothelial growth factor (VEGF) or its tyrosine kinases receptors [3,4]. In addition, other pro-angiogenic mediators or angiogenic-related pathways have been considered as potential therapeutic targets [5,6]. Unfortunately, although the treatments showed some form of activity, they did not significantly improve the prognosis of patients.

It is worth emphasizing that several different players are involved in cancer-associated angiogenesis. In recent times, exosomes have been proposed as an efficient cargo of pro-angiogenic mediators in cancer [7,8] although not in sarcomas. Exosomes are nanosized (30–100 nm) vesicles derived from the endosomal compartment of most cell types. They contain lipids, proteins, mRNAs, and microRNAs; providing a novel paracrine signaling mechanism for intercellular communications in physiological and pathological process [9,10]. Their role on cancer-associated angiogenesis has been described in melanoma [11], chronic myeloid leukemia [12], lung adenocarcinoma [13], glioblastoma [14] and renal cell carcinoma [15].

A common feature of solid tumors is their acidic microenvironment, resulting from an increased energetic metabolism [16]. In melanoma cells, variations in intracellular pH alters the lipid composition of the cell membrane and subsequently modulate both exosome release and uptake [17]. Recently, we have demonstrated that extracellular acidosis contributes to chemoresistance of OS cells [18] and promotes a metabolic reprogramming of OS cells thus contributing epigenetic stability in these cells [19]. However, the role of acidosis on extracellular nanovesicles release and activity has been poorly investigated so far in tumor angiogenesis.

In this study, we explored the hypothesis that osteosarcoma induced extracellular acidity may prompt extracellular nanovesicles (EVs) release and modify their pro-angiogenic activity. For this purpose, we purified and characterized EVs from OS cells maintained at acidic or neutral pH and showed, for the first time, that EVs contain functional pro-angiogenic miRNAs and proteins. This cargo provides an additional pathway for the cellular acquisition and dissemination of angiogenic traits and indicates extracellular nanovesicles as important mediators in the spread of angiogenesis activity in human osteosarcoma.

## 2. Results

### 2.1. Extracellular Nanovesicles Release Is Increased by Acidic pH

Extracellular nanovesicles isolated from OS cells displayed the exosome-like size range (30–100 nm) (Figure 1a), both when isolated from cells maintained at acidic (pH 6.5) or neutral (pH 7.4) conditions. Extracellular nanovesicles identity was confirmed by western blotting analysis for exosomal markers. As shown in Figure 1b; CD9, CD81, and CD63 were strongly enriched in EVs preparations compared to cell lysates and the heat-shock protein 70 (hsp70) was expressed in all samples. Ponceau S staining served as loading control (Appendix A). The release of EVs by OS cells was significantly increased at acidic pH (*p* = 0.002, *n* = 6) (Figure 1c). The doubling time of 143B cells maintained at acidic (pH 6.5) or neutral (pH 7.4) conditions was 22.0 h ± 0.9, and 18.8 h ± 0.6, respectively (*n* = 3). Acidosis slowed down cell growth and did not affect cell viability.

### 2.2. OS-Derived EVs Are Internalized by Endothelial Cells

To examine whether extracellular nanovesicles from OS cells were taken up by endothelial cells, PKH26 labelled EVs were incubated with HUVEC cells for 24 h and examined using fluorescence microscopy. As shown in Figure 2a, PKH26 signal was detected in the perinuclear region, suggesting the adsorption and internalization of EVs derived from OS cells maintained in acidic (pH 6.5) and neutral (pH 7.4) medium. No fluorescent signal was detected in the control.

### 2.3. OS-Derived EVs Did Not Affect Endothelial Cells Viability and Migration

HUVEC cells were exposed to OS-derived EVs to evaluate whether EVs could exert an effect on endothelial cell viability and migration. EVs, when isolated from OS cells maintained at either acidic or neutral pH, were not able to significantly affect cell viability (Figure 2b). The effects of OS-derived EVs on HUVEC motility were evaluated by the wound healing assay. After 24 h the migration of HUVEC cells was not influenced by the EVs treatment (Figure 2c).

### 2.4. OS-Derived EVs Promoted Endothelial Cells Tubulogenesis and Induced New Blood Vessel Growth In Vivo

OS-derived EVs induced HUVEC cells to form tube-like structures in vitro (Figure 3a). A significant increase (*p* = 0.034, *n* = 4) of total length of capillary tubes was identified for HUVEC cells treated with EVs derived from OS cells maintained both at pH 6.5 or pH 7.4 (Figure 3b). Additionally, a significant increase of the average total number of branch points was measured for HUVEC cells treated with OS-derived EVs (pH 6.5) when compared to the control (*p* = 0.05, *n* = 4) (Figure 3c). The pro-angiogenic activity of OS-derived EVs was confirmed in vivo by the chick chorioallantoic membrane (CAM) assay (Figure 3d). As shown in Figure 3e,f the length of vessels and the number of branch points was significantly higher in CAM treated with EVs derived from OS cells cultured at acidic pH (*p* = 0.018 and *p* = 0.0026, *n* = 7).

### 2.5. Angiogenesis-Related Proteins and miRNA Contained in OS-Derived EVs

The activity of OS-derived EVs could be ascribed to their cargo. Thus, we explored angiogenesis-related proteins expression in EVs by a membrane-based antibody array. A series of angiogenesis-related proteins (i.e., serpin E1, serpin F1, TIMP-1, thrombospondin-1, uPA, VEGF, pentraxin 3, PDGF-AA, angiopoietin-2, coagulation factor III, CD26, CD105, endostatin, endothelin-1, and HB-EGF) were found in OS-derived EVs (Figure 4a,b). No significant differences were found in angiogenic protein cargo between EVs isolated from OS cells maintained at acidic or neutral pH. To further explore EVs content, an angiogenesis-related miRNA profiling was carried out. We found detectable levels of miR-146a-5p, miR-10b-5p, miR-143-3p, miR-382-5p, miR-150-5p, miR-125b-5p, miR-27a-3p, miR-145-5p, miR-26a-5p, miR-93-5p, miR-21-5p, miR-92a-3p and miR-106a-5p (Figure 5) in OS-derived EVs. No difference in miRNA content was evidenced between EVs released at pH 7.4 or pH 6.5.

## 3. Discussion

The complex relationship existing among cancer cells and their microenvironment is still under investigation. Cell-to-cell communication is anything but simple, and secreted factors play a key role in the interaction among cells located far away [20]. In this context extracellular nanovesicles are emerging as important players of intercellular signaling mechanisms, including communication among cancer cells and their microenvironment [21].

The aim of this study was to investigate the role of EVs derived from osteosarcoma cells on angiogenesis and how extracellular acidity, generated by tumor metabolism, support EVs activity. To this purpose, we purified and characterized EVs from OS cells maintained at acidic or neutral pH and we showed, for the first time, that EVs contain functional pro-angiogenic miRNAs and proteins.

EVs are able to interact with target cells through various mechanisms. Since vesicles are membrane bound structures, it has been postulated that they may fuse and release their content in recipient cells. The mechanism is largely unknown; however, it has been proposed that the uptake may occur by receptor-mediated interactions or by direct fusion of EVs membrane with cell plasma membrane [22]. Furthermore, an acidic pH, which is typical of a hypoxic microenvironment of solid tumors; facilitates EVs membrane fusion and release of their content into the cytoplasm [23]. Ban J.J. et al. [24] demonstrated an increased production and stability of EVs at low pH. According to this, our results demonstrated that the acidic extracellular pH significantly increased the amount of EVs secreted from osteosarcoma cells. EVs from OS cells maintained at either acidic or neutral pH were morphologically homogeneous and displayed the exosome size range. Furthermore, OS-derived EVs were internalized by endothelial cells; without affecting cell viability.

Interestingly, we demonstrated that OS-derived EVs carry detectable amount of angiogenesis-related proteins. In particular, we identified VEGF, PDGF-AA, endostatin, ET-1, CD26, PAI-1, THBS1, uPA, ANG-2, TF3, TIMP-1, PEDF, PTX3, HB-EGF and CD105. All these proteins are involved in different steps of the angiogenic switch, that trigger the development of an infiltrating vascular network associated with tumor progression and metastasis formation.

In osteosarcoma, VEGF is a pro-angiogenic factor with a key role in pulmonary metastasis and poor prognosis [25,26]. The expression of PDGF-AA, endostatin, endothelin-1 and CD26 has been previously shown to be positively correlated with tumor progression in osteosarcoma [27,28,29,30] and the upregulation of serpin E1 and thrombospondin-1 in OS cells has been correlated with an increased risk of lung metastasis [31,32]. Additionally, Endo-Munoz L. et al., suggested an association between the metastatic behavior of OS and the activation of uPA axis carried by EVs at distant sites [33]. Among the other detected proteins, angiopoietin-2 expression was previously found to be correlated with tumor stage in osteosarcoma [34] and coagulation factor III (TF3) has been associated with survival and the regulation of tumor progression [35]. In osteosarcoma cells TIMP-1 has a dual influence on tumor progression; either beneficial by impairing angiogenesis or detrimental by favoring cancer cells growth or survival [36].

PEDF is an anti-angiogenic factor, as PEDF/VEGF ratio plays a crucial role in the process of vascular development and angiogenesis [37]. PEDF mRNA was found to be downregulated in exosomes isolated from poor responder OS patients [38]. According to this, here we found that OS-derived EVs contained nearly undetectable levels of this anti-angiogenic factor.

Interestingly, we identified other angiogenesis related proteins (i.e., PTX3, HB-EGF and endoglin) whose specific role in osteosarcoma has not been previously described.

PTX3 is a glycoprotein that plays an essential role in immune regulation, inflammation, apoptosis and vascular remodeling. Elevated PTX3 plasma levels were related to poor prognosis in various cancers although not sarcoma [39]. HB-EGF expression is significantly elevated in many human cancers and it has been associated with invasion, metastasis and resistance to chemotherapy [40]. Endoglin (CD105) is highly up-regulated in blood vessels of tissues where neovascularisation occurs, and its expression has been correlated with metastasis in breast cancer and colorectal tumor [41].

Extracellular vesicles represent an important mode of communication within the local tumor microenvironment and distant sites. The activity of EVs may also be ascribed to the presence of a specific repertory of RNAs in their cargo able to shift the balance to benefit pro-angiogenic signals.

In this study, the characterization of miRNA-cargo content within EVs, isolated from OS cells maintained at acidic or neutral pH, showed detectable amounts of angiogenesis-related miRNAs, including miR-10b-5p, miR-21-5p, miR-26a-5p, miR-27a-3p, miR-92a-3p, miR-93-5p, miR-106a-5p, miR-125b-5p, miR-143-3p, miR-145-5p, miR-146a-5p, miR-150-5p and miR-382-5p. The activity of these miRNAs is related to angiogenic pathways (Table 1). In particular, miR-93, miR-106a and miR150 were predicted to target VEGF [42,43].

In osteosarcoma, the expression of miR-10b and miR-146a has been related to tumor recurrence [62,63], and elevated levels of miR-21, miR-92a and miR-125b or low levels of miR-26a and miR-382 have been associated with poor prognosis [64,65,66]. Wang et al., have associated low levels of miR-150 with high tumor OS grade and poor response to chemotherapy [67].

In OS-derived EVs we identified miRNAs that have shown contrasting effects on osteosarcoma cell proliferation. Indeed, miR-93 has been associated with cell proliferation, while miR-143 has been related to an inhibitory activity on invasion [68,69].

Several attempts have been devoted to identify an osteosarcoma-specific microRNA or proteomic signature, thereby improving the existing tools for early diagnosis and disease staging by using liquid biopsies. The ideal biomarker must be accurate, sensitive and specific. In this view, miRNA enclosed in EVs should be more stable than circulating miRNA, as the lipid bilayer of exosomes makes a stable structure which is resistant to enzymatic degradation in blood [70]. On this basis a differential expression of miR-21, miR-27a, miR-143, miR-145, miR-146 and miR-382 has been found in exosomes isolated from serum of osteosarcoma patients with poor or good response to chemotherapy [38].

Our characterization of pro-angiogenic OS-derived EVs paves the way to investigate the role of additional extracellular vesicle miRNAs in OS diagnosis, progression and metastasis with the aim to improve the liquid biopsy application.

Moreover, we demonstrated that cancer acidity prompts EVs release. In this study, EVs were quantified as protein total mass. Cancer acidity may induce also differences in EVs number and size [71]. By evaluating the protein mass per single vesicle, we could have potentially used less vesicles in the pH 6.5 condition compared to pH 7.4. On this basis, the higher pro-angiogenic activity that we observed for OS-derived EVs isolated at acidic pH can be ascribed to EVs number or content.

Moreover, Parolini et al. demonstrated that, low extracellular pH (pH 6.0) affected the lipid composition of EVs secreted by melanoma cells. Authors suggested that this is likely responsible for the increased fusion efficiency of EVs secreted at acidic pH [17]. According to this we hypothesize that a similar mechanism could be responsible for a more efficient up take or release of content also for OS derived EVs at acidic pH.

The effect of the acidic microenvironment may be exploited to consider, from the pharmacological point of view, an adjuvant alkalinizing pharmacological approach, that could interfere with EVs release and activity, as suggested by Logozzi et al. [71], thus finally leading towards decreased OS progression.

## 4. Materials and Methods

### 4.1. Cell Culture

The human OS cell line 143B and the human primary human umbilical vein endothelial cells (HUVEC) were purchased from the American Type Culture Collection (ATCC, Manassas, VA, USA) and Promocell (VWR International Srl, Milan, Italy), respectively. The 143B cells were maintained in Iscove’s Modified Dulbecco’s Medium (IMDM, Invitrogen, Carlsbad, CA, USA), with a 25 mM D-Glucose, 4 mM L-glutamine, and 1 mM Sodium Pyruvate content and supplemented with 10% foetal bovine serum (FBS) (Sigma-Aldrich, Milan, Italy), penicillin (100 U/mL), and streptomycin (100 μg/mL) (Invitrogen). Only 143B cells in exponential growth phase were used. HUVEC cells were grown to confluence in Endothelial Cell Growth Medium (Promocell) plus Endothelial Cell Growth Supplement (Promocell) in tissue culture flasks precoated with 0.2% gelatine in water (Sigma-Aldrich). HUVEC cells were used up to passage 4. All cells were maintained at 37 °C in a humidified 5% CO_2_ atmosphere, and periodically tested for mycoplasma contamination.

### 4.2. Extracellular Nanovesicles (EVs) Isolation and Purification

The human OS cell line 143B cells were cultured until 70% confluence. Cells were washed with phosphate-buffered saline (PBS) and incubated for two consecutive periods (72 h and additional 18 h) with RPMI (Sigma-Aldrich) (pH 6.5 or 7.4) supplemented with 10% FBS depleted of exosomes (FDE) obtained via ultracentrifugation [72]. The specific pHe of the culture medium (6.5 or 7.4) was maintained by using different concentrations of sodium bicarbonate, according to the Henderson-Hasselbach equation. At the end-point of each experiment, the final pH in the supernatant was always measured with a digital pH-meter (6230N, Jenco, San Diego, CA, USA) to ascertain the maintenance of the pH value throughout the incubation time. Cell density and viability were assessed by the erythrosine B (Sigma-Aldrich) dye exclusion method [73]. Following collection of the supernatant from 143B cells grown on 15 petri dish (diameter 150 mm, 18 mL/petri), the EVs were concentrated by differential centrifugation: 500× *g* for 10 min (two times), 2000× *g* for 15 min (two times), and 10,000× *g* for 30 min (two times) at 4 °C to remove floating cells and cellular debris. The supernatant was then ultracentrifuged at 110,000× *g* for 1 h at 4 °C. The EVs pellet was resuspended in PBS and centrifuged at 110,000× *g* for 1 h at 4 °C (Beckman Coulter, Milan, Italy). The EVs pellet was resuspended in PBS and stored at −80 °C until use. EVs quantity was determined by the Bradford method (Bio-Rad, Milan, Italy). To check the effect of acidic pH on cell viability the doubling time of 143B cells, cultured at different pH, was evaluated. 143B cells were seeded at a density of 1 × 10^4^ cells/well (12 w plate) in RPMI medium supplemented with 10% FDE. After 24 h, cells were treated with RPMI (pH 6.5 or 7.4). Cells viability was assessed using the Erythrosine B assay at 24 h, 48 h and 72 h.

### 4.3. Electron Microscopy

EVs were resuspended in 2% paraformaldehyde (PFA) and loaded onto formvar-carbon coated grids. Next, EVs were fixed in 1% glutaraldehyde, washed, counterstained with a solution of uranyl oxalate, pH 7.0, and embedded in a mixture of 4% uranyl acetate and 2% methylcellulose before observation with a Zeiss-EM 109 transmission electron microscope (Zeiss, Jena, Germany).

### 4.4. Western Blot Analysis

EVs and cell pellets were treated with RIPA lysis buffer (25 mM Tris-HCl pH 7.6, 150 mM NaCl, 1% NP-40, 1% Na-deoxycholate, 0.1% SDS) and protease inhibitor cocktail (Roche, Milan, Italy) for 30 min at 4 °C. Nuclei and cell debris were removed by centrifugation. The protein concentration was determined using the Bradford assay (Bio-Rad). The total cellular proteins and exosomal proteins were resolved by 10% SDS-polyacrylamide gel and transferred to a nitrocellulose membrane (Thermo Fisher Scientific). The membrane was blocked with 5% dry milk (Thermo Fisher Scientific) in T-TBS (0.1 M Tris-HCl pH 8.0, 1.5 M NaCl and 1% Tween-20) for 1 h at room temperature. Subsequently, the membranes were incubated with rabbit polyclonal CD63, CD9, CD81, and hsp70 (EXOAB kit) (1:1000) (System Biosciences, Palo Alto, CA, USA) antibodies overnight at 4 °C. After vigorous washing in 0.05% Tween-20 in PBS, the membranes were incubated with the secondary antibody for 1 h at room temperature. Goat anti-rabbit antibody (EXOAB kit) (1:20,000) conjugated to horseradish peroxidase was diluted in 5% dry milk in T-TBS and used as secondary antibody. Immunocomplexes were detected with the ECL western blot analysis system (Euroclone, Milan, Italy). Reversible Ponceau S (Sigma Aldrich) staining was used to assess equal gel loading.

### 4.5. Extracellular Nanovesicles Labelling and Uptake

EVs were labelled using the PKH26 Red Fluorescent Cell Linker kit (Sigma-Aldrich) according to the manufacturer’s instructions with minor modifications. Two microgram (2 μg) of the PKH26 labelled EVs, or the same volume of the PKH26-PBS control, were resuspended in Endothelial Cell Growth Medium with 10% FDE and added to 1 × 10^4^ HUVEC cells maintained at 37 °C in a humidified atmosphere with 5% CO_2_. All samples were ultracentrifuged at 110,000× *g* for 1 h at 4 °C before being added to the cells and unincorporated dye from exosome labelling reactions was removed by using Exosome Spin Columns (MW 3000) (Thermo Fisher Scientific, Waltham, MA, USA) according to the manufacturer’s instructions. After 24 h of incubation, uptake was stopped by washing and fixation in 3.7% PFA for 10 min. Cells were then stained with a fluorescein isothiocyanate (FITC)-conjugated phalloidin (Sigma-Aldrich) and visualized with a Nikon Eclipse E800M fluorescence microscope (Nikon, Tokyo, Japan).

### 4.6. Endothelial Cells Viability Assay

Cell viability was evaluated by the acid phosphatase assay (Sigma-Aldrich). HUVEC cells were seeded in 96-well, flat-bottom tissue culture plates coated with 0.2% gelatin in water at a density of 1 × 10^4^ cells/well in Endothelial Cell Growth Medium plus Endothelial Cell Growth Supplement. After 24 h, cells were treated with 2–6 μg/w of OS-derived EVs (pH 6.5 or 7.4) or PBS (control), in the presence of Endothelial Cell Growth Medium + 1% FDE. At 48 h, the cells were washed and incubated at 37 °C with 100 μL of buffer containing 0.1 M sodium acetate pH 5.0, 0.1% Triton X-100, and 5 mM p-nitrophenyl phosphate. After 2 h, the reaction was stopped with the addition of 10 μL of 1 N NaOH, and color development was assayed at 405 nm using a microplate reader (Tecan Infinite F200pro, Männedorf, Switzerland). Data are reported as cell survival in respect to untreated cells (set = 1). The experiment was performed three times in triplicate.

### 4.7. Endothelial Cells Migration Assay

HUVEC cells were seeded in 24-well, flat-bottom tissue culture plates coated with 0.2% gelatin in water at a density of 2 × 10^4^ cells/well in Endothelial Cell Growth Medium plus Endothelial Cell Growth Supplement. Cell monolayers at confluence were scraped in a straight line to create a ‘scratch’ with a p100 pipet tip. Medium was changed to remove floating cells and cells were incubated with 2 μg/w of OS-derived EVs (pH 6.5 or 7.4) or PBS (control), in the presence of Endothelial Cell Growth Medium + 1% FDE. After 16 h, the images acquired for each sample were photographed under a phase-contrast microscope. To obtain the same field during the image acquisition, markings were created as reference points close to the scratch. The rate of migration was measured by quantifying the wound closure by Image J software [74]. The experiment was performed three times in duplicate.

### 4.8. In Vitro Tubulogenesis Assay

HUVEC cells were seeded in 48-well, flat-bottom tissue culture plates coated with 125 μL/w of matrigel growth factor reduced (BD Biosciences, Erembodegem, Belgium) at a density of 4 × 10^4^ cells/well. Cells were incubated with 5 μg/w of OS-derived EVs (pH 6.5 or 7.4) or PBS (control) diluted in Endothelial Cell Growth Medium + 1% FDE. Tube-like formation was documented after 5 h with photomicrographs taken at 10× magnification. Capillary tube-like length was quantified using the Image J software.

### 4.9. Chick Chorioallantoic Membrane (CAM) Angiogenesis Assay

Fertilized specific pathogen-free chicken eggs (Charles River Laboratories Italia, Milan, Italy) were incubated at 37 °C in a humidified atmosphere (at > 60% relative humidity) in an incubator with a moving tray, rotating the eggs continuously. On development day three, the eggs were opened under laminar airflow. The level of the CAM was lowered by removing two mL of albumen through a 21-gauge needle inserted at the tip of the egg. Tegaderm film (3M Italia, Milan, Italy) was applied at the upper side of the shell in order to prevent spilling of shell particles onto the CAM while cutting the window. Then, a window of approximately 1 cm^2^ was opened in the egg shell by using a pair of sterile sharp-pointed surgical scissors. The window was sealed with Tegaderm film to prevent dehydration and eggs were incubated at 37 °C on 90% (*v*/*v*) relative humidity. On chick embryonic development day 6, the eggs were inoculated under laminar airflow. Tegaderm film was removed and a silicon O-ring was placed on the top of growing CAM. OS-derived EVs (4 μg/egg in 50 μL PBS), both from pH 7.4 and 6.5 culture conditions, were mixed with 50 μL/w matrigel growth factor reduced on melting ice. The mixed solution was then put inside the silicon O-ring over the CAM. PBS was used as control. The window was closed with Tegaderm film and the eggs were returned for incubation till day 9. In ovo images were acquired with the stereomicroscope SMZ18 (Nikon Instruments, Kanagawa, Japan) equipped with a DS-Ri2 Digital Microscope Camera Digital Photo color camera DFC (Nikon) at 50× magnification. After image acquisition, the internal area of the O-ring was analysed by Image J software and the growth of the vessels was measured by means of its length and number of vessels junctions [75]. All the in vivo CAM experiments were performed according to the Directive 2010/63/EU of the European Parliament and of the council of 22 September 2010 on the protection of animals used for scientific purposes. Accordingly, in ovo experiments did not require any special additional allowance as long as the embryos were sacrificed before hatching, as was done in this study.

### 4.10. Expression Profiles of Angiogenesis-Related Proteins

EVs were lysed in PBS containing 1% Triton X-100 and protease inhibitor cocktail (Roche, Milan, Italy) for 30 min at 4 °C. Nuclei and cell debris were removed by centrifugation. The protein concentration was determined using the BCA Protein Assay Kit (Thermo Scientific) and 150 μg of EVs proteins were analysed by Human Angiogenesis Array Kit (R&D Systems, Minneapolis, MA, USA) according to the manufacturer instructions. Positive reference spots are used to orientate the arrays and to identify the exposure that exhibits a high signal to noise ratio. This array is designed for the detection of activin A, a disintegrin and metalloproteinase with thrombospondin motifs 1 (ADAMTS-1), angiogenin, angiopoietin-1, angiopoietin-2 (ANG-2), angiostatin, amphiregulin, artemin, coagulation factor III (TF3), chemokine ligand 16 (CXCL16), dipeptidyl peptidase-4 (CD26), epidermal growth factor (EGF), endocrine gland derived vascular endothelial growth factor (EG-VEGF), endoglin (CD105), endostatin, endothelin-1 (ET-1), fibroblast growth factor (FGF) acidic, FGF basic, FGF-4, FGF-7, glial cell-derived neurotrophic factor (GDNF), granulocyte-macrophage colony-stimulating factor (GM-CSF), heparin-binding EGF-like growth factor (HB-EGF), hepatocyte growth factor (HGF), insulin-like growth factor-binding protein 1 (IGFBP-1), IGFBP-2, IGFBP-3, interleukin-1 beta (IL-1β), IL-8, transforming growth factor (TGF- β1), leptin, monocyte chemoattractant protein-1 (MCP-1), macrophage inflammatory protein (MIP-1α), matrix metalloproteinase (MMP-8), MMP-9, neuregulin (NRG1- β1), pentraxin 3 (PTX3), platelet-derived endothelial cell growth factor (PD-ECGF), PDGF-AA, PDGF-AB, pershephin, platelet factor 4, placental growth factor (PIGF), prolactin, serpin E1 (PAI-1), serpin F1 (PEDF), serpin B5, tissue inhibitor of metalloproteinase-1 (TIMP-1), TIMP-4, thrombospondin -1 (THBS1), thrombospondin-2, urokinase-type plasminogen activator (uPA), vasohibin, VEGF and VEGF-C. The histogram profiles for detected proteins were generated by quantifying the mean spot pixel densities from the array membrane by using Quantity One software (Biorad Laboratories Headquarters, Hercules, CA, USA). The experiment was performed two times in duplicate.

### 4.11. Expression of Angiogenesis-Related miRNA

Total RNA was isolated from EVs by using miRCURY™ RNA Isolation kit (Exiqon, Vedbaek, Denmark) according to the manufacturer instruction. UniSp6 (Exiqon) was added as spike-in control into the lysis solution. Total RNA was quantified using Nano Drop ND-1000 spectrophotometer (Thermo Fisher Scientific Waltham, MA, USA) and reverse-transcribed with miRCURY LNA Universal kit (Exiqon). An angiogenesis-related miRNA profiling was carried out by using miRCURY SYBR Green PCR Kit (Exiqon) and miRNA primer sets (Exiqon). miRNA name and target sequences were described in Table 2. Real-time PCR was performed by using the CFX96 Touch™ Real-Time PCR Detection System (Bio-Rad, Segrate (MI), Italy). The amplification curves were analysed by using the Bio-Rad CFX software, both for determination of Cq (by the 2nd derivative method) and for melting curve analysis. Only miRNA with Cq < 35 were considered detected. The data were analyzed by the ΔΔCq method and considering miR-23a-3p and miR-16-5p as references [76]. The experiment was performed three times in duplicate.

### 4.12. Statistical Analysis

Statistical analysis was performed using the Graph Pad Prism 7.04 software for Windows (Graph Pad Software, La Jolla, CA, USA). Results were reported as mean ± standard deviation or mean ± standard error and the differences were analyzed using non-parametric Mann–Whitney test for the difference between groups. Only *p* < 0.05 were considered significant.

## 5. Conclusions

This study provides evidence that extracellular nanovesicles shed by human osteosarcoma cells act as carriers of active angiogenic stimuli, both proteins and miRNAs, which are able to promote endothelial cell functions relevant to angiogenesis. Finally, our findings suggest that the acidic extracellular pH of the microenvironment plays an important role in the modulation of angiogenesis through the increased release of EVs from osteosarcoma cells.

## Figures and Tables

**Figure 1 cancers-11-00779-f001:**
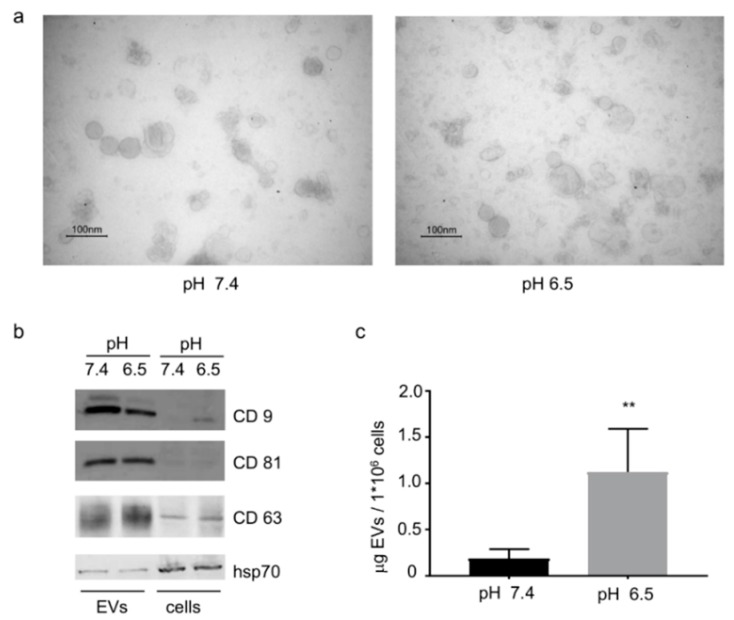
Osteosarcoma (OS)-derived extracellular nanovesicles (EVs) characterization. (**a**) Representative transmission electron microscopy images of EVs, isolated from medium conditioned by 143B cells, maintained at pH 7.4 or pH 6.5. (**b**) EVs enrichment was assessed by western blot analysis for the expression of the specific exosomal marker CD9, CD81, CD63 and hsp70. (**c**) The release of EVs by OS cells was quantified by protein assay and normalized on 1 × 10^6^ viable cells (mean ± SD, *n* = 6, Mann–Whitney test, ** *p* value < 0.01).

**Figure 2 cancers-11-00779-f002:**
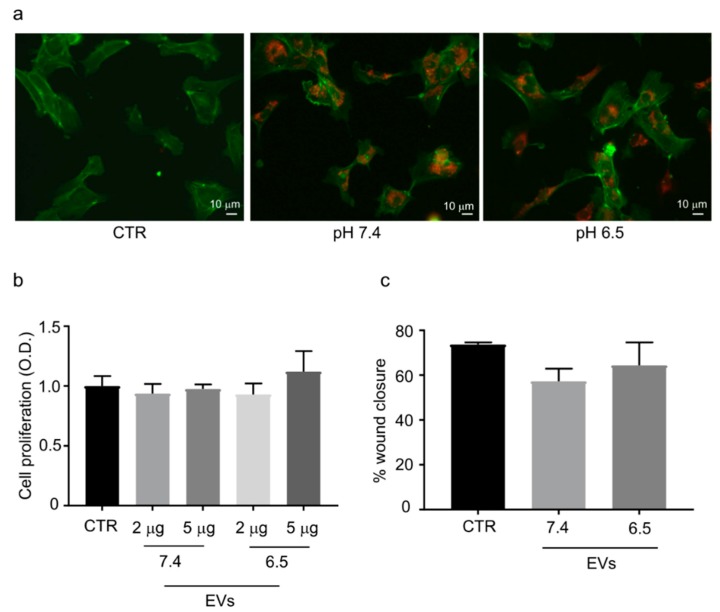
OS-derived EVs were uptaken by endothelial cells, and they did not affect cell viability and migration. (**a**) The uptake of the fluorescently labelled (red) was evident in HUVEC cells after 24 h of incubation. No stain was revealed in the PBS negative control condition (CTR). Actin filaments were stained with a FITC-conjugated phalloidin (green). Representative images. Scale bar = 10 µm (20× objective; 200 magnification). (**b**) HUVEC cells viability was not affected by EVs treatment. (**c**) HUVEC cells migration was not affected by the treatment with OS–derived EVs. Graphs represent triplicate biological repeats and are displayed as mean ± SD, Mann–Whitney test, CTR: HUVEC cells not treated with EVs.

**Figure 3 cancers-11-00779-f003:**
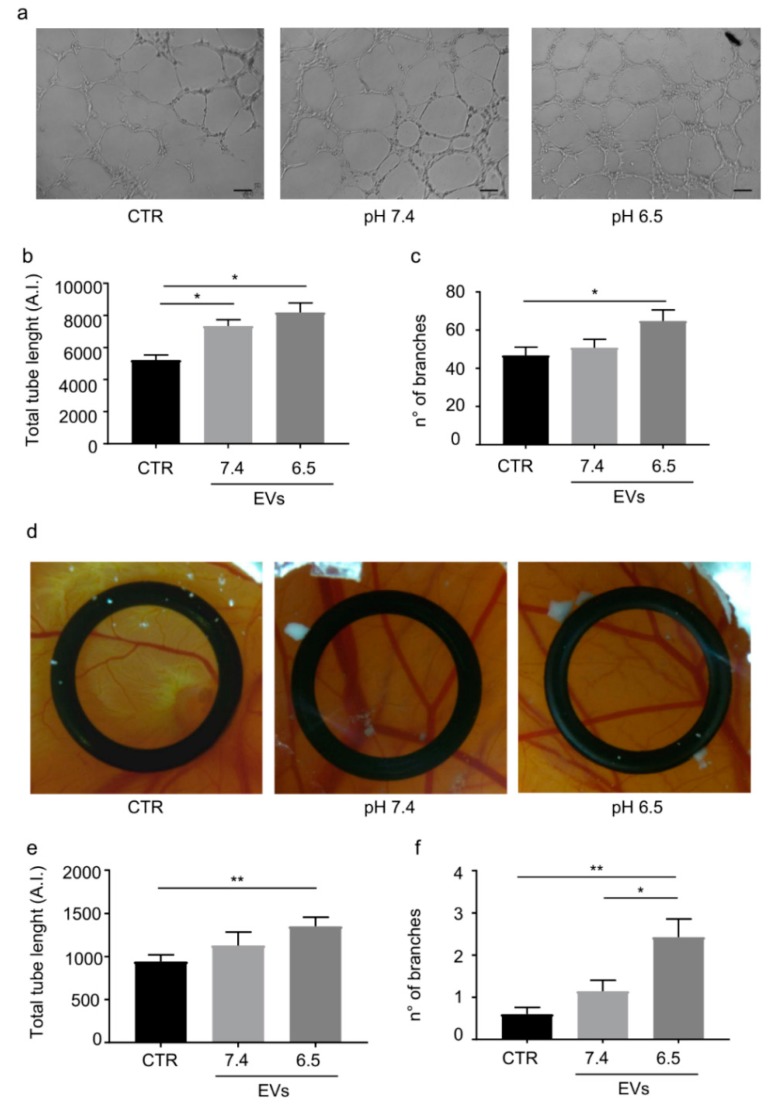
OS-derived EVs promoted significantly tube formation by HUVEC cells in vitro and new blood vessel growth on the chorioallantoic membrane (CAM) vascular bed. (**a**) Representative images of HUVEC cells on matrigel in presence of OS-derived EVs (scale bar = 100 µm) (10× objective; 100 magnification). (**b**) The treatment with EVs derived from OS maintained at acidic and neutral pH significantly increased the total tube length in vitro (AI: arbitrary unit; mean ± SE, *p* = 0.034 for CTR versus pH 7.4 and for CTR vs. pH 6.5, *n* = 4, Mann–Whitney test). (**c**) The number of branches was increased by the treatment with EVs isolated by OS maintained at acidic pH (mean ± SE, *p* = 0.05, *n* = 4, Mann–Whitney test). (**d**) CAM: representative images of new blood vessel growth after the treatment with OS-derived EVs. (**e**) Quantitative analysis of the total tube length revealed that EVs derived from OS cells maintained at acidic pH induced vessel growth (AI: arbitrary unit) (mean ± SE, *p* = 0.0018, *n* = 7, Mann–Whitney test). (**f**) The number of vessels junction is significantly increased by EVs obtained from OS cells maintained at acidic pH (mean ± SE, *p* = 0.0026, *n* = 7, Mann–Whitney test). * *p* value < 0.05; ** *p* value < 0.01.

**Figure 4 cancers-11-00779-f004:**
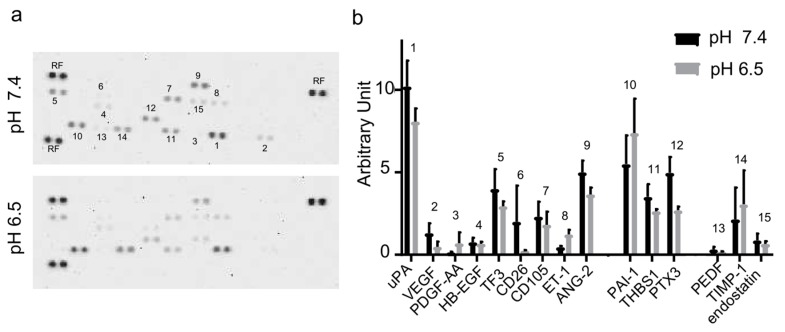
EVs cargo: angiogenesis-related proteins. (**a**) Representative images of the positive spots associated with the angiogenesis-related proteins detected by the human angiogenesis array (RF: reference spot) (**b**) Densitometric quantification of angiogenesis array signal (mean ± SE, Mann–Whitney test). The experiment was performed two times in duplicate. (1–9: pro-angiogenic factors; 10–12: pro/anti-angiogenic factors; 13–15: anti-angiogenic factors).

**Figure 5 cancers-11-00779-f005:**
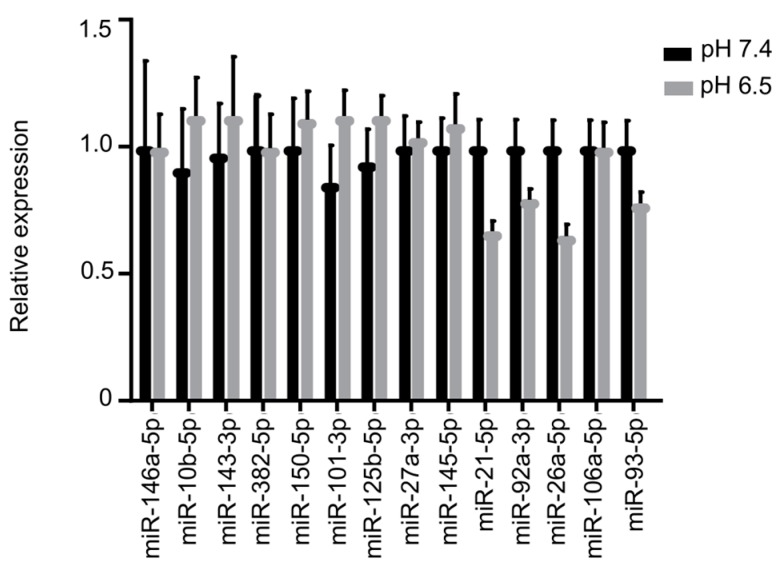
EVs cargo: angiogenesis-related miRNA. Real Time PCR analysis of the expression of angiogenesis-related miRNA in OS-derived EVs maintained at acidic or neutral pH (mean ± SE, *n* = 3, Mann–Whitney test).

**Table 1 cancers-11-00779-t001:** miRNA identified in OS-derived EVs, specific targets and function in angiogenesis.

microRNA	Target	Function	References
miR-10b-5p	HOXD10, FLT1	Positive regulation of VEGF receptor signaling pathway	[44,45]
miR-21-5p	PTEN	Increasing VEGF production	[46,47]
miR-26a-5p	AKT and ERK1/2 Nogo-B receptor SMAD-1 and SMAD-4	Up-regulation of HIF-1 α expression Decreasing the VEGF-induced phosphorylation of the endothelial nitric oxide synthase	[48,49]
miR-27a-3p	Sprouty2 and Sema6A	Increase endothelial cells mediated angiogenesis	[50]
miR-92a-3p	VHL gene ITGA5 and MEK4	Stabilization of HIF-1 α, thus promoting VEGF transcription. Down-regulation of HGF secretion	[51,52]
miR-93a-5p	Integrin β8, LATS2, PTEN, VEGF	Inhibition of EPLIN expression in endothelial cells.	[42,45,53,54]
miR-106a-5p	VEGF	Anti-angiogenic	[42]
miR-125b-5p	HER2, HER3	Decrease of ERBB2 and VEGF expression Inhibits translation of VE-cadherin	[45,55]
miR-143-3p	CAMK1D	Increases tube formation by endothelial cells	[56]
miR-145-5p	CAMK1D, P70S6K1	Increases tube formation by endothelial cells Inhibition tumor angiogenesis	[45,56]
miR-146a-5p	Smad4, HAb18G	Increases the expression of VEGF Promoting PDGFRA expression Downregulates VEGF	[57,58,59]
miR-150-5p	ING4, VEGF	Up-regulation the secretion of VEGF Negative regulator of VEGF A	[43,60]
miR-382-5p	PTEN	Increases vascular endothelial cell proliferation, migration and tube formation	[61]

**Table 2 cancers-11-00779-t002:** miRNA name and target sequences.

miRNA Name	Target Sequence
hsa-miR-101-3p	UACAGUACUGUGAUAACUGAA
hsa-miR-106a-5p	AAAAGUGCUUACAGUGCAGGUAG
hsa-miR-10b-5p	UACCCUGUAGAACCGAAUUUGUG
hsa-miR-125b-5p	UCCCUGAGACCCUAACUUGUGA
hsa-miR-143-3p	UGAGAUGAAGCACUGUAGCUC
hsa-miR-145-5p	GUCCAGUUUUCCCAGGAAUCCCU
hsa-miR-146a-5p	UGAGAACUGAAUUCCAUGGGUU
hsa-miR-150-5p	UCUCCCAACCCUUGUACCAGUG
hsa-miR-16-5p	UAGCAGCACGUAAAUAUUGGCG
hsa-miR-210-3p	CUGUGCGUGUGACAGCGGCUGA
hsa-miR-214-3p	ACAGCAGGCACAGACAGGCAGU
hsa-miR-21-5p	UAGCUUAUCAGACUGAUGUUGA
hsa-miR-23a-3p	AUCACAUUGCCAGGGAUUUCC
hsa-miR-26a-5p	UUCAAGUAAUCCAGGAUAGGCU
hsa-miR-27a-3p	UUCACAGUGGCUAAGUUCCGC
hsa-miR-296-5p	AGGGCCCCCCCUCAAUCCUGU
hsa-miR-29b-3p	UAGCACCAUUUGAAAUCAGUGUU
hsa-miR-34a-5p	UGGCAGUGUCUUAGCUGGUUGU
hsa-miR-382-5p	GAAGUUGUUCGUGGUGGAUUCG
hsa-miR-92a-3p	UAUUGCACUUGUCCCGGCCUGU
hsa-miR-93-5p	CAAAGUGCUGUUCGUGCAGGUAG

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
