# Peer review of "Extracellular Nanovesicles Secreted by Human Osteosarcoma Cells Promote Angiogenesis"

_cancers, 2019, doi:10.3390/cancers11060779_

Round 1
Reviewer 1 Report
The revised version of the manuscript addresses the main points raised in the previous review and is acceptable for publication.
Author Response
Thank you.
Reviewer 2 Report
The aim of this study was to investigate the role of EVs derived from osteosarcoma cells on angiogenesis and whether extracellular acidity, generated by tumor metabolism, could influence EVs activity.
The are of research in this area is important, but the manuscript need major revision.
- In figure 2 panel (a) need to add the magnification and a bar, the same for panel (d )
The quality of the microphotography of panel (d) is bad. The author should change these microphotographies.
Also it is important to add the magnification used in figure 3 panel a
When there is an in-vitro experiments to evaluate angiogenesis or vasculogenesis, the authors should do a different type of evaluation, because the evaluation that the authors have done is for in-vivo experiments. I recommend to review other report published in order to do a good evaluation.
The results obtained with respect to EVs-cargo angiogenesis -related proteins, how many experiments were done? The authors must explain. On the other hand, it is important to explain what means the differences found in same of those protein with statistical difference with pH
Also, the author should explain in more detail the statistical difference they found in miRs levels of miR-21-5p, miR-92a-3p, miR-26a-5p and miR-93-5p with the different pH and also to add the number of experiments they did
Author Response
The aim of this study was to investigate the role of EVs derived from osteosarcoma cells on angiogenesis and whether extracellular acidity, generated by tumor metabolism, could influence EVs activity.
The are of research in this area is important, but the manuscript need major revision.
- In figure 2 panel (a) need to add the magnification and a bar, the same for panel (d)
According to reviewer, we added the magnification in the figure 2 legend (panel a and panel d). Moreover, we added the length in the bar.
The quality of the microphotography of panel (d) is bad. The author should change these microphotographies.
According to reviewer, we changed the microphotographies of the Figure 2 panel d.
Also it is important to add the magnification used in figure 3 panel a
According to reviewer, we added the magnification in the figure 3 legend (panel a).
When there is an in-vitro experiments to evaluate angiogenesis or vasculogenesis, the authors should do a different type of evaluation, because the evaluation that the authors have done is for in-vivo experiments. I recommend to review other report published in order to do a good evaluation.
Different processes are involved in angiogenesis. Among these, endothelial cells migration, proliferation, differentiation and structural rearrangement are central. The assays that we performed described different aspects of angiogenetic process. Wound healing migration assay highlights the motile activity of cells and describes a multi-step process involving spreading, proliferation and migration of cells. Tubulogenesis assay, that simulate the formation of capillary-like tubules, is described as representative of the later stages of angiogenesis (differentiation) (Staton CA et al, 2004). According to De Cicco-Skinner KL et al, the tube formation assay is a fast, reproducible and sensitive method for in vitro measurement of angiogenesis. The choice to measure tube length and the number of branches in tube formation assay in vitro is commonly used (Delle Monache S et al, 2013; Sun H et al, 2018). Since in vitro angiogenesis assays only recapitulate a few steps of the angiogenic process and, though very reproducible, they are not necessarily an accurate reflection of blood vessel formation (Irvin MW et al, 2014), we also performed also an in vivo assay (i.e. chick chorioallantoic membrane (CAM) assay).
Staton CA, Stribbling SM, Tazzyman S, Hughes R, Brown NJ, Lewis CE. Current methods for assaying angiogenesis in vitro and in vivo. Int J Exp Pathol. 2004 Oct;85(5):233-48
De Cicco-Skinner KL, Henry GH, Cataisson C, Tabib T, Gwilliam JC, Watson NJ, Bullwinkle EM, Falkenburg L, O'Neill RC, Morin A, Wiest JS. Endothelial cell tube formation assay for the in vitro study of angiogenesis J Vis Exp. 2014 Sep 1;(91): e51312. doi: 10.3791/51312.
Delle Monache S, Angelucci A, Sanità P, Iorio R, Bennato F, Mancini F, Gualtieri G, Colonna RC. Inhibition of angiogenesis mediated by extremely low-frequency magnetic fields (ELF-MFs). PLoS One. 2013 Nov 14;8(11): e79309. doi: 10.1371/journal.pone.0079309.
Sun H, Wang S, Song M. Long non‑coding RNA SENCR alleviates the inhibitory effects of rapamycin on human umbilical vein endothelial cells. Mol Med Rep. 2018 Aug;18(2):1405-1414. doi: 10.3892/mmr.2018.9094.
Irvin MW, Zijlstra A, Wikswo JP, Pozzi A. Techniques and assays for the study of angiogenesis. Exp Biol Med (Maywood). 2014 Nov;239(11):1476-88. doi: 10.1177/1535370214529386.
The results obtained with respect to EVs-cargo angiogenesis -related proteins, how many experiments were done? The authors must explain. On the other hand, it is important to explain what means the differences found in same of those protein with statistical difference with pH
Also, the author should explain in more detail the statistical difference they found in miRs levels of miR-21-5p, miR-92a-3p, miR-26a-5p and miR-93-5p with the different pH and also to add the number of experiments they did.
EVs cargo angiogenesis-related proteins was evaluated two times in duplicate, as reported in the Material and Methods (paragraph 4.10). We reported this information also in the legend of Fig. 4 EVs cargo: angiogenesis-related proteins. The experiment for the evaluation of miRNA levels was performed three times in duplicate. This information has been reported in Material and Methods (paragraph 4.11) and in Fig 5 legend.
Our results showed that acidosis increases OS-EVs release but does not impact quantitatively their angiogenic cargo both as proteins and miRNAs. Even if, according to the reviewer, the levels of miR-21-5p, miR-92a-3p, miR-26a-5p and miR-93-5p were lower in EVs isolated at acidic pH when compared to pH 7.4, this difference was not statistically significant. We hypothesized that the higher pro-angiogenic activity that we observed for OS-derived EVs isolated at acidic pH can be ascribed to increased fusion efficiency of EVs secreted at acidic pH.
Reviewer 3 Report
revised form accepted
Author Response
Thank you.
Round 2
Reviewer 2 Report
The authors must repet the migration assay and to have a better photography
with the quality of that results the manuscript can not be published
The image is not good in figure 2 Panel d. For this reason I inssit that author must do the experiments again and to take e good picture. If it is not possible the authors must to take this panel from figure 2.
Author Response
The authors must repeat the migration assay and to have a better photography with the quality of that results the manuscript can not be published. The image is not good in figure 2 Panel d. For this reason, I inssit that author must do the experiments again and to take e good picture. If it is not possible the authors must to take this panel from figure 2.
According to reviewer, we removed the images of wound-healing assay from Figure 2, panel d. The effect of OS-derived EVs on endothelial cell migration is represented by the graph in Figure 2 panel c. Results showed that the migration of HUVEC cells was not affected by the EVs treatment.
This manuscript is a resubmission of an earlier submission. The following is a list of the peer review reports and author responses from that submission.
Round 1
Reviewer 1 Report
The manuscript by Perut and colleagues explores the potential role of pH in osteosarcoma(OS)-derived exosomes effects in angiogenesis. Although being a relevant topic, the paper still misses important data on the relation between exosomes cargo, pH and angiogenesis.
Specifically, in the experimental design the authors didn't explore the potential impact of pH in the number and average protein cargo of exosomes produced by the OS model here studied. This is exemplified in the reference #71 (Logozzi et al, Cancers, 2018), where it was shown that changes in pH impact both exosomes #/ml and exosomes protein cargo. Specifically, together with an increase of ~40x in protein mass, there was also an increase of ~3x in exosomes number in OS under low pH. Based on these date, it is possible to estimate that there was an increase of ~10X in the protein mass per exosome in low pH. By normalizing exosomes preparation only by ug/cells the authors may not only be losing important information on the potential impact of exosomes packaging of biomolecules and overall shedding of vesicles, but also may be utilizing different number of vesicles in each of the experiments presented in the manuscript. Considering that most analyses presented in the paper by Perut et al were performed by using equal amounts of ug of exosomes, the authors may have potentially used less vesicles in the pH6.5 condition when compared to the pH7.4 one. If this is the case, it could significantly impact the results and conclusions of the experiments here presented. Therefore, it would be highly recommendable to utilize equal number of vesicles (measured by, for instance, NTA), instead of equal protein amount in the different experimental conditions.
In addition, as there are no experimental evidence of transference of miRNAS and proteins to endothelial cells, it would be highly recommendable that authors tone down the conclusions presented in the second paragraph of the discussion. Alternatively, the targeting of specific miRNA and/or protein species in OS cells and exosomes followed by analysis of potential impact in angiogenesis would greatly improve the significance of the paper.
Reviewer 2 Report
The manuscript title “Exosome-like nanovesicles secreted by human osteosarcoma cells promote angiogenesis” has interesting contributions to the research area.
The aim of this study was to investigate the role of EVs derived from osteosarcoma (OS) cells on angiogenesis and whether extracellular acidity, generated by the tumor metabolism, could influence EVs activity.
The authors purified and characterized EVs from OS cells maintained at either acidic or neutral pH and they evaluated the ability of exosomes to induce angiogenesis assessed in vitro by endothelial cell tube formation and in vivo using chicken chorioallantoic membrane.
Questions:
1.- Why did they not use a control for the western-blots experiments?
2.- The authors must explain in each figure legends the representation of the results ( mean +/- standar error or standard deviation) and also the statistic test used
3.- It is difficult to see the significant differences in figure 3c and figure 3e. Please explain which was the representation of the results and the statistical test.
This study provides evidence that extracellular nanovesicles secreted by human osteosarcoma cells act as carriers of active angiogenic stimuli and also the importance of the tumour metabolism.
Reviewer 3 Report
In their study, Perut et al. purified and characterized extracellular vesicles from osteosarcoma cell line 143B, produced in acidic condition. They show an increase of EVs production when cells are cultured at pH 6,5. They investigated the angiogenic potential of OS-derived Evs and demonstrate that OS-EVs were pro-angiogenic a) in in vitro model of HUVEC tubulogenesis on Matrigel, independently of pH used for EVs production b) in in vivo model of CAM assay only when EVs produced in acid condition. Qualitative analysis of EVs cargo related to angiogenesis reveals the presence of several proteins and miR, but with no significant quantitative differences between EVs produced in neutral or acidic pH.
Overall, this paper is interesting and well-written but I have some concern, questions and comments that need to be convinsingly addressed.
First of all, this paper does not meet the Minimal Informations for study of extracellular vesicles, as stated by the International Society for Extracellular Vesicles (2014, updated in 2018, Thery et al, Journal of Extracellular Vesicles).
· The nomenclature should be corrected, as the term “exosomes” is not appropriate in this study.
· As far as studies EVs are isolated from conditioned medium, the appropriate negative control has to be complete medium that has not been conditioned by cells, but still processed in the same way as conditioned medium. It does not seem to be the case in this work as the control (CTR) is indicated as PBS, with no more details. Authors should clarify this issue.
· Even if the quantification of EVs released by OS cells has been normalized on viable cells, authors do not explicit the effect (if any) of acidosis on OS cell viability. This issue should be mentioned in the text.
In Figure 2b, primary endothelial HUVEC cells are cultured 48 h in endothelial medium without supplements and with very low serum (0,1% serum FDE). Then a proliferation assay is performed. In these experimental conditions, HUVEC cells are not able to proliferate, as these are “starving” conditions! HUVEC cannot cope with such low serum for more than 24h. They will engage either their apoptotic or their senescent program.
In Figure 2b, images of the wound should be shown.
In the tumor, endothelial cells are also facing an acidic environment, coming from the tumor metabolism but also from their own metabolism, as endothelial cells are highly glycolytic. Have the authors check OS-EVs on endothelial cells in an acidic medium?
How authors explain that OS EVs has not effect on HUVEC proliferation and migration but stimulate these cells in Angiogenesis assay?
I do not understand the statistics and the p values given for figure 3. Are P values the same for pH 7.4 and pH 6.5 in Figure 3b? Please indicate to which p value * or ** referred to in the figure 3.
Based on the findings of this work, acidosis increases OS-EVs release but does not impact quantitatively their angiogenic cargo (proteins and miR). Then, how do authors explain the angiogenic effect of pH 6.5 EVs nor 7.4 on CAM Assay, as the same amount of EVs (4 µg/egg) was loaded in each case?
In the Angiogenesis Array, what is the internal loading protein control used? How authors normalize for loading in order to compare arrays membrane? This point is not mentioned in the Mat/Meth Section.
In Figure 4b, pro-angiogeninic vs. anti-angiogenic actors could be indicated and classified on the graph as so. This would help the reader.
I have an important concern about the tumoral specificity of EVs in this study. As OS are from mesenchymal stromal cell origin, have the authors compare EVs derived from OS and derived from mesenchymal stromal cells MSC differentiated into osteoblasts? Is there any osteosarcoma specificity in the angiogenesis-related profile of EVs presented here?